# Towards Global Health Equity: A Prototype for Standardizing Patient Satisfaction Measurement in Alignment with the United Nations’ Sustainable Development Goals

**DOI:** 10.3390/healthcare13070697

**Published:** 2025-03-21

**Authors:** Mathew P. Morico, Samuel E. Neher

**Affiliations:** McGovern Medical School at UTHealth Houston, Houston, TX 77030, USA; samuel.e.neher@uth.tmc.edu

**Keywords:** patient satisfaction, global survey, healthcare quality

## Abstract

**Background/Objectives:** Global patient satisfaction is vital for healthcare quality, yet no survey enables effective cross-country comparisons. Existing tools are limited in scope, focusing on aspects like Patient-Reported Experience Measures, Patient-Reported Outcome Measures, or systemic factors within individual settings. This study introduces the Morico International Healthcare Satisfaction Survey prototype to facilitate global comparisons, aligning with the United Nations’ Sustainable Development Goals. **Methods:** We developed the prototype by reviewing existing surveys to identify relevant question formats, thematic focuses, and distribution methods. Surveys were chosen for their relevance and established use in healthcare settings. Our analysis highlighted common elements and gaps, guiding the design of a comprehensive tool that integrates Patient-Reported Experience Measures, Patient-Reported Outcome Measures, and systemic factors for global comparisons. **Results:** Existing surveys varied significantly in length and focus, revealing a lack of standardization. Best practices emphasized concise, clear surveys with standardized responses and online distribution. Our prototype includes 25 questions across eight sections, adapted for global use and broader healthcare systems. It provides a comprehensive framework for international comparisons. **Conclusions:** No standardized survey exists for cross-country healthcare comparisons. The Morico International Healthcare Satisfaction Survey prototype addresses this gap, paving the way for future refinement and implementation to enhance global healthcare quality and policy development.

## 1. Introduction

As clinical outcomes continue to improve and become more consistent worldwide, greater attention has shifted toward patient satisfaction as a key component of healthcare quality assessment [1,2]. Defined as a patient’s response to various aspects of their service experience, patient satisfaction not only holds inherent value, but it also offers tangible benefits to both patients and providers, justifying its role in healthcare quality assessments [2]. Studies have consistently shown that higher patient satisfaction correlates with better health outcomes, improved adherence to treatment, and fewer malpractice claims [3,4]. Consequently, being able to measure and compare patient satisfaction globally is essential for benchmarking healthcare quality and addressing disparities, particularly in underserved regions. This aligns with the United Nations Sustainable Development Goals (SDGs), specifically SDG 3: Good Health and Well-Being and SDG 10: Reduced Inequalities [5]. Moreover, SDG 17: Partnerships for the Goals underscores the importance of international collaboration in strengthening healthcare systems [5]. By benchmarking patient satisfaction across nations, we can identify and address inequities in healthcare delivery, advancing these global objectives and fostering more equitable healthcare systems. However, a standardized, easily accessible, and comprehensive tool for comparing patient satisfaction across healthcare systems worldwide does not exist. In its absence, policymakers and researchers lack a consistent framework for evaluating healthcare quality on a global scale, limiting opportunities for cross-country benchmarking and improvement [6].

Various models have been used to conceptualize the measurement of patient satisfaction, but tools generally fall into three categories: Patient-Reported Experience Measures (PREMs), which assess perceived service quality, communication, and interactions with service providers; Patient-Reported Outcome Measures (PROMs), which evaluate symptom relief and functional improvement; and system-wide surveys, such as public opinion polls, which gauge broader healthcare factors like accessibility, affordability, and overall trust in the system [7,8,9]. While each of these survey types provides valuable insights, they are rarely integrated, making it difficult to capture a country’s overall healthcare satisfaction [10]. For example, a country may excel in PREMs, with strong patient-provider interactions, score poorly in PROMs, with low post-surgical improvement rates, and have mixed perceptions of the healthcare system as a whole due to broader systemic issues. Without a unified approach that incorporates all three dimensions—experiences, outcomes, and system-wide factors—assessments remain fragmented, preventing meaningful comparisons of healthcare quality across countries.

For instance, widely used PREM-based tools such as the Hospital and Consumer Assessment of Healthcare Providers and Systems (HCAHPS) from the United States, the National Health Service Inpatient Survey (NHSIP) from the United Kingdom, and the Patient Experience Questionnaire (PEQ) from Norway assess only patient experiences in specific care settings without evaluating health outcomes [6]. Conversely, PROM-focused tools like the EQ-5D from the Netherlands, the SF-36 from the United States, and the AQoL-8D from Australia measure health-related quality of life but exclude experience-based factors [11,12,13]. System-wide surveys such as the LatinoBarómetro from Chile, the National Health Survey from Japan, and the National Family Health Survey from India, measure broader issues such as public health, policy, and accessibility but tend to lack in addressing personal medical encounters [14,15,16]. While these tools effectively assess individual dimensions of satisfaction, they are often limited to specific contexts and fail to provide a holistic view of healthcare satisfaction, which is inherently multifaceted. Furthermore, many of these surveys are designed for internal use and benchmarking rather than public accessibility, and their antiquated distribution and analysis methods—often relying on mail-in responses instead of online formats—further complicate cross-country comparisons and limit their utility as a standardized global measure [17,18].

To address these gaps, this study proposes the Morico International Healthcare Satisfaction Survey (MIHSS), a prototype that integrates PREMs, PROMs, and system-wide metrics into a single, comprehensive tool. By incorporating key elements from existing surveys, the MIHSS aims to establish a standardized framework for measuring healthcare satisfaction globally. Additionally, its intention for online distribution prioritizes accessibility, ensuring that data can be used for both public and policy-driven decision-making. By providing a holistic assessment that captures patient experiences, health outcomes, and systemic factors, the MIHSS could facilitate international benchmarking, inform policy decisions, and contribute to achieving the SDGs by promoting equitable, high-quality healthcare worldwide.

This article outlines the development of the MIHSS and explores its potential to improve global healthcare quality assessments by addressing the longstanding challenges of standardization, accessibility, and comprehensiveness.

## 2. Materials and Methods

To develop the Morico International Healthcare Satisfaction Survey (MIHSS), we conducted a structured review of existing patient satisfaction surveys to identify common question formats, thematic focuses, and distribution methods. This review aimed to inform the MIHSS design by examining how widely used tools capture PREMs, PROMs, and systemic factors.

Surveys were selected based on four criteria: (1) tools that explicitly measured patient satisfaction within healthcare settings, (2) established and reputable tools, defined as those widely cited, frequently adapted in multiple studies, or developed by recognized organizations such as national health agencies, (3) publicly available tools to allow for full content analysis, and (4) diversity in country representation to assess global interpretations of patient satisfaction.

A search was conducted using PubMed, Google Scholar, and general web searches to identify relevant surveys. When surveys were originally developed in another language, official English adaptations were used for analysis. One primary reviewer identified and gathered survey instruments, while a secondary reviewer independently confirmed their inclusion and analysis. Each survey was examined for structure, response formats, and thematic content to identify common design elements and areas where existing tools may be lacking. A summary of the key characteristics of the reviewed surveys is presented in Table 1.

Additionally, a complementary literature review was conducted to identify best practices in survey design, particularly for patient satisfaction surveys in healthcare settings requiring cultural and linguistic adaptation. PubMed searches, aided by artificial intelligence tools, were used to refine the search scope and identify relevant studies. This review focused on key methodological considerations, including survey readability, response format selection, and distribution strategies, to ensure that the MIHSS would be both comprehensive and broadly applicable across diverse populations.

## 3. Results

### 3.1. Findings from Survey and Literature Review

Our analysis of existing patient satisfaction surveys revealed several key findings. First, Likert-scale response formats were widely used and accepted as a standardized way to assess patient perceptions. Second, survey length varied significantly, ranging from as few as 5 questions to over 80, highlighting the lack of a consistent standard for survey length. Third, while surveys covered a broad range of topics, there was notable overlap in question content within similar types of surveys—for example, PREM surveys often included comparable themes and question structures. These findings informed the selection of key sections for inclusion in the MIHSS.

The literature review on best practices in survey design further guided the MIHSS’s structure and implementation strategy. Studies emphasized that shorter surveys with clear and concise language improve completability across different education and linguistic levels and increase response rates [26,27]. Additionally, standardizing response options using Likert scales with particularly five numerical values enhances accuracy and comparability [28]. Finally, online distribution was identified as an effective method for maximizing participation and streamlining data collection and analysis [29,30].

By integrating insights from both the survey and literature reviews, we developed the MIHSS as a structured, standardized tool for measuring patient satisfaction on a global scale.

### 3.2. Survey Structure

The MIHSS prototype consists of 25 questions divided into eight sections: one demographics section and seven core sections assessing satisfaction in specific areas of healthcare. Each core section contains two to three key questions that best represent its category. The demographics section uses multiple-choice questions, while all other sections employ a five-point Likert scale. The survey was created using Google Forms for online completion.

### 3.3. Adaptation and Broadening for Global Applicability

To maximize its relevance across diverse healthcare systems, the MIHSS was systematically adapted from other reference surveys in two key ways. First, the scope of the survey was broadened to assess healthcare satisfaction at a national level rather than being limited to specific settings. For example, questions were framed as “in your country” rather than “at clinic X” or “at hospital Y” to allow for more generalizable responses. Second, terminology was adjusted to be more inclusive of different healthcare structures and professions, replacing role-specific terms such as “nurses” or “doctors” with “healthcare professionals.” This ensured that the survey remained applicable across varying healthcare models.

### 3.4. Survey Sections

The MIHSS includes sections that capture essential dimensions of healthcare satisfaction, based on a review of existing satisfaction surveys. These sections are designed to provide a well-rounded assessment of healthcare quality, integrating PREMs, PROMs, and systemic factors.

Demographics: Collects background information such as country of residence, age, gender, location, and healthcare coverage, allowing for stratified analysis across different population groups.Personal Outcome Measures (PROMs): Assesses perceived health improvements and ability to perform daily activities post-care, offering a generalizable measure of health-related quality of life.Quality of Care (PREMs): Focuses on the effectiveness, safety, and overall experience of medical treatments, including provider competence and frequency of medical errors, to evaluate care delivery.Provider Engagement (PREMs): Evaluates communication, respect, and shared decision making from healthcare professionals, ensuring that interpersonal aspects of care are adequately represented.Costs and Value (systemic factors): Assesses affordability, perceived value, and fairness of healthcare costs, capturing economic factors influencing satisfaction.Accessibility (systemic factors): Examines ease of scheduling appointments, wait times, and availability of healthcare services to ensure timely care.Public Health (systemic factors): Expands beyond individual experiences to evaluate broader healthcare system functions, such as public health campaigns, emergency preparedness, and access to resources.Overall Satisfaction: Reflects general experiences with healthcare systems, summarizing respondents’ satisfaction and likelihood of recommending the healthcare system.

### 3.5. Considerations in Survey Design

The MIHSS was designed to be clear, concise, and accessible while providing a comprehensive assessment of healthcare satisfaction. A key challenge was balancing specificity and generalizability—capturing essential satisfaction metrics while ensuring broad applicability across diverse populations.

To maximize accessibility, the survey uses simple language, a standardized five-point Likert scale, and a consistent format, making it easy to understand and compare responses globally. The online structure allows for a seamless, sequential progression through sections, enhancing clarity and user engagement.

Designed as both an assessment tool and a reflective exercise, the survey moves from personal health outcomes to individual healthcare experiences, then to broader systemic factors, before concluding with an overall satisfaction assessment. This structured progression encourages respondents to reflect on their own experiences before evaluating the healthcare system as a whole.

By integrating these elements, the MIHSS provides a standardized, globally applicable framework for measuring healthcare satisfaction while remaining accessible to diverse populations.

### 3.6. The MIHSS Tool

The complete MIHSS prototype, including all survey items in its online format is provided in the Appendix A for reference. This is the original English version, as translations have not yet been conducted.

## 4. Discussion

While useful in their respective contexts, existing patient satisfaction tools are not adequate for cross-country comparisons. Most are designed for national or institutional use, often focusing on a single dimension—either PREMs, PROMs, or systemic factors—without integrating these perspectives into a comprehensive framework. Additionally, variations in survey length, format, and distribution methods make direct international comparisons challenging. This study highlights these limitations and introduces the MIHSS as a potential solution. By incorporating PREMs, PROMs, and systemic factors into a single tool, the MIHSS aims to provide a standardized and globally applicable measure of patient satisfaction.

While the MIHSS represents an important step toward addressing this gap, it remains a prototype that will benefit from further refinement through peer review and real-world application. Its design balances conciseness with comprehensiveness, but certain areas may require further expansion. Future iterations could explore extended versions for deeper insights while ensuring clarity and accessibility remain central to its structure. Refinements in wording and phrasing will also be essential to optimize clarity and applicability across diverse healthcare contexts.

A key challenge moving forward is ensuring accurate translations and cultural adaptability for international deployment, which can be a very time consuming and difficult process [31]. Linguistic nuances and cultural differences can significantly impact respondents’ interpretations of survey items, necessitating careful adaptation to maintain validity across diverse populations [32]. Additionally, considerations around data governance must be addressed, including how responses will be processed, stored, and safeguarded to ensure ethical use and security. Establishing a clear framework for data ownership and accessibility will be critical in maintaining transparency and trust.

The next phase of this research will involve piloting the MIHSS in one or more countries to assess feasibility, reliability, and effectiveness. Implementation strategies must consider best practices for survey distribution, response rates, and data management. A long-term goal is to create a global patient satisfaction database that is not only useful for policymakers but also accessible to the public. A publicly available, interactive visualization—such as a global heatmap of patient satisfaction—could provide valuable insights and foster greater accountability in healthcare systems worldwide.

## 5. Conclusions

As it stands, no standardized and comprehensive survey exists for cross-country comparisons. The MIHSS takes the first step toward narrowing this gap and advancing global healthcare quality benchmarking to align with the SDGs. While there is a long road ahead in refining and expanding the tool, this prototype provides a crucial foundation for addressing an important issue. Future efforts will focus on refining its methodology, ensuring cultural adaptability, and developing scalable implementation strategies. Despite the challenges ahead, the MIHSS serves as a meaningful first step in establishing a global framework for measuring patient satisfaction, paving the way for improved healthcare evaluation and policy development.

## Figures and Tables

**Table 1 healthcare-13-00697-t001:** Representative Healthcare Satisfaction Surveys from Various Countries.

Tool	Country of Development	Focus	Setting	Format	Length	Distribution Method
Hospital Consumer Assessment of Healthcare Providers and Systems (HCAHPS) [19,20]	United States	PREMs	Inpatient care	Likert scales and multiple-choice	32 questions	Mail, telephone, mail with telephone follow-up, active interaction voice recognition (IVR)
National Health Service Adult Inpatient Survey 2024 (NHSIP) [21,22]	United Kingdom	PREMs	Adult inpatient care	Likert scales, multiple-choice, open-ended	59 questions	Mail, online
Patient Experience Questionnaire (PEQ) [23]	Norway	PREMs	Outpatient care	Likert Scales	18 questions	Mail
EuroQol 5-Dimension (EQ-5D) [24]	Netherlands	PROMs	Nonspecific	Likert Scales	5 questions	Paper, online
Short Form Survey-36 (SF-36) [12]	United States	PROMs	Nonspecific	Likert Scales	36 questions	Varies
Assessment of Quality of Life 8-Dimension (AQoL-8D) [13]	Australia	PROMs	Nonspecific	Likert Scales	35 questions	Varies
LatinoBarómetro [14]	Chile	System-Wide Factors	General Healthcare	Likert Scales, multiple-choice, open-ended	80+ total questions, 5-10 healthcare related questions	Face-to-face interviews
Survey on Healthcare in Japan (SHJ) [15]	Japan	System-Wide Factors	General healthcare	Multiple-choice	22 questions	Online
National Family Health Survey [16,25]	India	System-Wide Factors	General Healthcare	Likert Scales, multiple-choice, open-ended. Four distinct questionnaires: Household, Women’s, Men’s, and Biomarker.	Varies	Computer Assisted Personal Interview (CAPI)

## Data Availability

The data that support the findings of this study are available from the corresponding author, upon reasonable request.

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
