# Peer review of "Towards Global Health Equity: A Prototype for Standardizing Patient Satisfaction Measurement in Alignment with the United Nations’ Sustainable Development Goals"

_healthcare, 2025, doi:10.3390/healthcare13070697_

Round 1

Reviewer 1 Report

Comments and Suggestions for Authors

Thank you for the opportunity to review this manuscript. The methodology employed in this study contains significant omissions which undermines the validity and reliability of the results. Additionally, the presentation of the results is suboptimal and necessitates revision. The discussion section contains results and requires revision and further information to be added to ensure in depth discussion of the results. Consequently, this article requires a major review to be considered suitable for publication.

Comments on the Quality of English Language

Could be improved in sections 

Reviewer 2 Report

Comments and Suggestions for Authors

Dear Author(s),

I had the pleasure of reviewing your manuscript ‘Toward Global Health Equity: Standardizing Patient Satisfaction Measurement for Better Comparisons in Alignment with the UN’s SDG’s’. Measuring patient satisfaction is a very important, relevant and pertinent topic. The authors provide an interesting review of existing measurements, highlighting their strengths and weaknesses.   

Although I think the study is interesting and worth publishing, I would like to suggest some improvements that can be summarized as follows:

Main concerns

  1. Abstract:
  2. Authors should avoid using acronyms in the abstract. Thus, I suggest the replacement of the acronyms SDGs, PREMs, and PROMs by their meaning.
  3. I suggest removing from the Results the reference to the Japan Survey on Healthcare because it constitutes a particular case in the more general analysis that is described.
  4. Results:
  5. Based on an article I found, and the authors cited [7] I think the authors missed referring to the Picker Patient Experience Questionnaire (PPE)-15 disseminated through USA and Europe.
  6. I would like to see in Table 1 examples of countries who used each survey metric, The authors said primary used in… but it would be interesting to see more detail.
  7. Discussion:
  8. Authors said that the development of a standardized survey must address cultural sensitivity and patient characteristics. Also, the authors provide references for patients’ characteristics, I think it could be improved. Also examples and references should be provided for cultural sensitivity.

In conclusion, I would like to say that although I find the topic interesting, the work is a little poor. In fact, it adds little or nothing to other work already published. I would like to see a proposal for a globally applicable questionnaire, even if it is a prototype to be tested. This could be an important contribution from which new studies could emerge. I therefore invite the authors to propose a questionnaire that measures the multiple components of patient experience. Essentially, a questionnaire that combines the three dimensions presented in the article - constructs from PREM, PROM and systemic metrics. The Japanese model seemed the closest to this purpose, so the authors could try to improve it. 

 Good luck!

Round 2

Reviewer 1 Report

Comments and Suggestions for Authors

Manuscript Title: Toward Global Health Equity: Standardizing Patient Satisfaction Measurement for Better Comparisons in Alignment with the UN’s SDG’s

Overall:

Thank you for the opportunity to review this manuscript. Now renamed as “Towards Global Health Equity: A Prototype for Standardizing 2 Patient Satisfaction Measurement in Alignment with the 3 United Nations’ Sustainable Development Goals”. The authors have made substantial revisions to the manuscript, effectively redefining and refining the content they are presenting. I commend them for the thoroughness and rigor demonstrated in this extensive revision process. However, there are a few minor adjustments that should be made before the manuscript is ready for publication. These are outlined below.

Abstract

Satisfactory changes

Introduction:

Lines 45-57 please provide references to support these statements.

Lines 76-80 57 please provide references to support these statements.

Material and Methods

While the SF36 was replaced in this study with the AQoL-8D, it would be beneficial to include a statement to support this choice given the SF36 is so widely published.

Results:

Thank you for revising the table the information is clearer with these headings.

The revisions and change of focus of this section is commendable.

When the reader clicks on the link to the survey via https://forms.gle/TkgLE3JgMYpj2cnGA , does it open in languages other than English? If not, please add this in here and discuss in a line or two.

Discussion

This section has been revised to address the changes earlier in the manuscript.

Conclusion:

This section has been updated to reflect the revisions made earlier in the manuscript.

Author Response

Please see updated manuscript attached, with revisions in RED. Below are point-by-point responses to your comments. Thank you for your time once again!

Comments 1: Introduction: Lines 45-57 please provide references to support these statements. Lines 76-80 57 please provide references to support these statements.

Response 1: Thank you for pointing this out. We have included 5 references to support the claims from lines 45-57 (now 47-58 in the new manuscript). These claims we make are novel, but based on prior work done outlined in these studies. We have also included 2 references to support the claims from lines 76-80 (now 78-82 in the new manuscript).

Comments 2: Material and Methods: While the SF36 was replaced in this study with the AQoL-8D, it would be beneficial to include a statement to support this choice given the SF36 is so widely published.

Response 2: After reviewing the updated manuscript, we decided to re-include the SF36. You are correct that it is widely published. Initially, we omitted it to allow for broader country inclusion (since we already discussed HCAHPS from the U.S.), but given the prominence of the SF36, we felt its inclusion was more beneficial. Please refer to line 70 and the updated table for its re-inclusion. Additionally, its inclusion brings balance to the table, as it now features three surveys each for PREMs, PROMS, and system-wide surveys. Thank you for pointing this out!

Comments 3: Results: When the reader clicks on the link to the survey via https://forms.gle/TkgLE3JgMYpj2cnGA , does it open in languages other than English? If not, please add this in here and discuss in a line or two.

Response 3: We have addressed this issue in updated lines 214-215 and 261. 

Reviewer 2 Report

Comments and Suggestions for Authors

Dear Authors,

I appreciate your work with the revision. 

You should have highlighted the changes in the text in a different colour to make them easier to see. I think that's a requirement of the journal. 

I have nothing more to ask.

Best regards,

Author Response

Comments 1: 

I appreciate your work with the revision. 

You should have highlighted the changes in the text in a different colour to make them easier to see. I think that's a requirement of the journal. 

I have nothing more to ask.

Best regards

Response 1: Thank you for taking the time to review this updated version. We are glad you accept the improvement. We apologize for not using a different color for changes in this update. Our revisions were drastic, so much of it was rewritten and the color change would have been about half of the manuscript, so we opted to do a comment-by-comment document instead. Thank you for your understanding!